# Hybrid Materials with Antimicrobial Properties Based on Hyperbranched Polyaminopropylalkoxysiloxanes Embedded with Ag Nanoparticles

**DOI:** 10.3390/pharmaceutics15030809

**Published:** 2023-03-02

**Authors:** Alexander Vasil’kov, Dmitry Migulin, Alexander Naumkin, Ilya Volkov, Ivan Butenko, Alexandre Golub, Vera Sadykova, Aziz Muzafarov

**Affiliations:** 1A. N. Nesmeyanov Institute of Organoelement Compounds, RAS, 119991 Moscow, Russia; 2Enikolopov Institute of Synthetic Polymeric Materials, RAS, 117393 Moscow, Russia; 3G. F. Gause Institute of New Antibiotics, 119021 Moscow, Russia

**Keywords:** silver nanoparticles, antimicrobial activity, polyaminopropylsiloxanes, hyperbranched molecular architectures, metal–vapor synthesis, X-ray photoelectron spectroscopy

## Abstract

New hybrid materials based on Ag nanoparticles stabilized by a polyaminopropylalkoxysiloxane hyperbranched polymer matrix were prepared. The Ag nanoparticles were synthesized in 2-propanol by metal vapor synthesis (MVS) and incorporated into the polymer matrix using metal-containing organosol. MVS is based on the interaction of extremely reactive atomic metals formed by evaporation in high vacuum (10^−4^–10^−5^ Torr) with organic substances during their co-condensation on the cooled walls of a reaction vessel. Polyaminopropylsiloxanes with hyperbranched molecular architectures were obtained in the process of heterofunctional polycondensation of the corresponding AB_2_-type monosodiumoxoorganodialkoxysilanes derived from the commercially available aminopropyltrialkoxysilanes. The nanocomposites were characterized using transmission (TEM) and scanning (SEM) electron microscopy, X-ray photoelectron spectroscopy (XPS), powder X-ray diffraction (PXRD) and Fourier-transform infrared spectroscopy (FTIR). TEM images show that Ag nanoparticles stabilized in the polymer matrix have an average size of 5.3 nm. In the Ag-containing composite, the metal nanoparticles have a “core-shell” structure, in which the “core” and “shell” represent the M^0^ and M^δ+^ states, respectively. Nanocomposites based on silver nanoparticles stabilized with amine-containing polyorganosiloxane polymers showed antimicrobial activity against *Bacillus subtilis* and *Escherichia coli*.

## 1. Introduction

The increase in the number of strains of microorganisms that are resistant to most antibiotics makes it useless to use previously developed and effectively used antibiotics and constantly stimulates the search for new antibacterial drugs and materials. The search for alternatives to the antibiotics used goes in different directions, one of which is to assess the possibility of using biologically active metal nanoparticles and their oxides [1,2].

The use of metal nanoparticles with high biological activity for the production of medical materials is one of the most relevant and intensively developed scientific directions. Silver nanoparticles have long attracted the attention of researchers from the field of medicine and healthcare. A significant number of products containing silver nanoparticles have been released: surgical instruments and dressings, clothing, cosmetics, food packaging, etc. [3].

However, there is a problem associated with the degradation and deterioration of products based on metal nanoparticles—this is the lack of stability and the tendency of nanoscale metal particles to aggregation, arising from strong interactions between these nanoparticles caused by their high surface energy.

The creation of nanocomposites based on metal nanoparticles and a polymer matrix stabilizing them can solve this issue [4]. Such composites consist of two or more phases that differ significantly in their properties. Various compounds were used to stabilize metal nanoparticles: organic and inorganic ligands, small organic molecules, and linear and dendritic polymers [5].

Organosilicon polymers have attracted attention with properties such as environmental friendliness, hydrophobicity, abrasion resistance, physiological inertia, nontoxicity and so on [6,7]. The biocompatibility and inertness of polymers allow them to be used for soft tissue implants or heart valve replacement [8], in the production of contact lenses and in the treatment of corneal injuries [9].

It is known that one of the most effective modification methods is the modification of the surface of metal nanoparticles with alkoxysilanes containing various functional groups [10]. Therefore, Demin et al. used trimethoxysilane to stabilize nanoparticles based on Fe_3_O_4_ and Fe_2_O_3_ (3-aminopropyl) [11].

Hyperbranched polymers composed of dendritic, linear and terminal units can be produced in a one-pot reaction and a number of different hyperbranched type of dendritic polymers such as polyamidoamines, poly(ethylene imine), polyglycerol, polyester, poly(acryl amide), etc. have been widely used for the stabilization of various metal nanoparticles [12,13].

Besides organic polymers, heteroorganic functional polyorganosiloxanes with hyperbranched molecular structures have recently become an object of interest in terms of their capability towards coordination and stabilization of metal nanoparticles.

The class of functional polyorganosiloxane hyperbranched structures represents objects of molecular topology with a branched surface, a large number of terminal functional groups. The special chemical nature of the siloxane backbone is responsible for low viscosity, outstanding wetting and film-forming properties, oxidative and thermal stability over a wide temperature range, physiological inertness, and UV and visible light transparency. In such structures, silicon atoms form all branch centres. They are effective templates for the coordination and encapsulation of metal nanoparticles [14,15,16,17,18,19]. The creation of organometallic nanocomposites based on them is a promising direction for many high-tech applications in the field of catalysis, biomedicine, photonics, etc. [20].

Fabrics are a necessary component of medical supplies. At the same time, they serve as one of the main means of spreading pathogenic microorganisms, since they do not have bactericidal properties. In this regard, the creation of modern technologies for the production of antibacterial materials is an urgent task [21,22,23].

One of the possibilities for giving antibacterial properties to fabrics is the application of thin composite films containing silver on their surface. Silver has been effectively incorporated into fabrics as an antimicrobial agent due to its strong inhibitory and antimicrobial effects on a broad spectrum of bacteria, fungi and viruses with low toxicity to humans. In recent years, studies have been carried out on the formation of polymer coatings containing metallic silver nanoparticles by metal deposition or co-deposition of metal and matrix material under low-pressure plasma [24,25,26,27,28,29].

Methods of tissue functionalization with antimicrobial agents are in high demand in medical textiles. However, the development of such systems to control the release of antimicrobial agents is hindered by the absence of functional groups on the tissue. The addition of organomarites to stabilize the silver nanoparticles on the tissue also contributes to their protection from further oxidation and controls their release. In this regard, the use of superbranched organosilicon polymers is an urgent task.

In the currently known methods for obtaining metal nanoparticles in a biopolymer matrix, as a rule, procedures for the chemical reduction of metal salts in a biopolymer solution are used. These methods have a number of serious limitations that significantly complicate the use of the obtained materials for biomedical purposes. The following stand out among them: the presence of a significant amount of impurities of surfactants and residues of synthesis products, as well as the complexity of controlling the completeness of metal reduction.

One of the promising methods for the synthesis of metal nanoparticles is metal–vapour synthesis (MVS). The method is based on simultaneous processes of evaporation and condensation of metal and organic ligand on the reactor walls cooled by liquid nitrogen under vacuum conditions of 10^−4^–10^−5^ Torr. MVS does not introduce restrictions when choosing a metal or a combination of metals and can be carried out for almost any combination of organic reagent–metal. Since the 60s of the last century, when the MVS first demonstrated its extensive synthetic capabilities, a significant amount of research has been carried out to study the reactivity of atoms and clusters of metals under the conditions of this method. Initially, MVS was used in organometallic chemistry to prepare metal complexes that are stable at low temperatures in argon atmosphere. However, in the context of growing interest in nanotechnology and ultradisperse systems, the method has received a new round of development.

To date, nanoparticles of metals of gold, silver, copper, nickel, cobalt, zinc and many other metals “solvated” by various types of solvents have been obtained by MVS [30,31,32]. The particle size can be controlled by varying the nature of the metal–organic reagent pair, along with some other synthesis parameters (pressure, evaporation rate, etc.).

The advantages of this method of obtaining metal nanoparticles include: the absence of synthesis by-products during the formation of metal nanoparticles, which is especially important for obtaining biomedical materials; the possibility of obtaining nanoparticles of various metals, including those with pronounced antimicrobial activity and/or magnetic properties; and ease of modification of various types of carriers, including biopolymer matrices with the purpose of giving them new functional properties.

Nanocomposites based on natural polymers containing Ag nanoparticles have shown high antibacterial activity [33,34,35,36].

The fundamental possibility of using the MVS to obtain medical supplies was demonstrated in [37,38]. Unlike most methods of obtaining nanoparticles, the MVS is completely environmentally friendly and can be easily integrated into various technological cycles.

The effectiveness of the use of MVS for the creation of new hybrid metal-containing materials with fungicidal activity against two types of pathogenic microorganisms, *Sclerotium rolfsii* and *Rhizoctonia solani*, was demonstrated [39]. Biocidal activity of the prolonged action of silver nanoparticles obtained by the MVS against *E. coli* and *S. aureus* was shown, as well [40].

Composites based on chitosan and nanoparticles of metals Au, Co, Ni and Cu were obtained using the method of metal–vapor synthesis in [41,42,43]. Not only were the bactericidal and bacteriostatic activities of composites against *E. coli*, *S. aureus* and *S. enterica* evaluated, but also toxicological tests were carried out on laboratory mice. It has been shown that composites obtained using MVS, when used in certain concentration ranges, do not exhibit toxicity in the short-term (3 days) and long-term (14 days) perspective in experimental animals [42].

In this regard, the use of MVS, which can be easily integrated into technological processes, can be effective for obtaining medical materials.

The task set within the framework of this study is focused on the creation of a new generation of medical products based on hybrid systems containing metal nanoparticles for potential clinical use. To stabilize metallic silver nanoparticles and obtain functional nanocomposite coatings with antibacterial properties, previously developed amino-containing polyorganosiloxane structures with a controlled hyperbranched molecular architecture were used [16].

## 2. Results and Discussion

Hybrid materials based on organosilicon polymers of branched structure, modified with nanoparticles of biologically active metals, can be promising in the creation of new medical materials due to the fact that these polymers, along with thermal stability and resistance to oxidation, are highly compatible with many materials. An introduction of metal nanoparticles contributes to the acquisition of a composite complex of functional properties: antibacterial, magnetic, catalytic and others.

Coordinatively active hyperbranched polyaminopropylethoxysiloxanes (HBPAPES), with the flexible branched molecular architecture of the polyethoxysiloxane backbone and functional amino-groups, were synthesized according to the previously described process [16] of controlled polycondensation of AB_2_-type monosodium salts of an aminopropyldiethoxysilane monomer, bearing two types of chemically independent functional groups in one molecule (Figure 1 and Figure 2)

Initial AB_2_-type sodiumoxy-(3-aminopropyl)diethoxysilane was synthesized in the reaction between the equimolar amounts of commercially available aminopropyltriethoxysilane and sodium hydroxide (Figure 1).

The quantitative yields and purity of the product were determined directly by means of ^1^H, ^13^C, ^15^N, ^29^Si NMR and mass spectroscopy methods (see Appendix A).

The following polycondensation step was carried out through the neutralization of the AB_2_-type sodiumoxy-(3-aminopropyl)diethoxysilane with an equimolar amount of acetic acid, resulting in the preparation of the corresponding hyperbranched polyorganoalkoxysiloxane (Figure 2).

^1^H and ^29^Si NMR spectroscopies were used for the characterization of the molecular structures of the thus obtained functional polyorganosiloxane (see Appendix A).

The ^29^Si NMR method with the addition of the paramagnetic chromium(III) acetylacetonate, serving as a relaxation accelerator [44], allowed us to obtain data on the degree of branching (DB) in the synthesized polyaminopropylethoxysiloxane. DB was found to be equal to 0.43, confirming that the obtained polymer is characterized with a relatively high DB, peculiar for the polymers with hyperbranched molecular structures. 

For the first time, nanocomposites were obtained based on a biocompatible polymer matrix—cotton, Ag nanoparticles obtained by MVS using isopropanol as a dispersion medium and organosilicon nanogel Ag NPs/HBPAPES/cotton. Figure 3 shows the synthesis scheme of the obtained nanocomposites.

With the help of modern research methods, the properties of new nanocomposite materials were studied.

The FTIR spectral analysis was conducted in order to determine the molecular interaction between HBPAPES and silver nanoparticles. Figure 4 shows the IR spectra of pure HBPAPES, characterised with the absorption bands at 3355 cm^−1^ (O–H and N–H stretching vibrations), 2863 cm^−1^ (C–H stretching vibrations) and 1623 cm^−1^ (N–H bending vibrations).

The FTIR spectra of Ag NPs/HBPAPES shows weakening and broadening of O–H and N–H stretching vibrations at 3355 cm^−1^ and shifting of N–H bending vibrations to 1579 cm^−1^, resulting from the binding of polymers’ -NH_2_ functional groups with silver nanoparticles. 

Figure 5 shows TEM micrographs of Ag NPs in a bright field (**a**) and a histogram of the distribution of Ag nanoparticles from the Ag NPs/2-propanol /HBPAPES system (**b**).

Analysis of micrographs showed that the particles are spherical with an average size of 5.3 ± 0.2 nm. It was observed that organosilicon nanogel stabilizes silver nanoparticles well in isopropanol organosole.

To assess the morphology and composition of the Ag NPs/HBPAPES/cotton nanocomposite, SEM was used, and its X-ray energy dispersion spectrum was recorded (Figure 6).

SEM showed that the metal nanoparticles have a wide size distribution after 2-propanol removal. The metal particles are aggregated into bunch-like objects in which smaller metal nanoparticles are combined (Figure 7).

The powdered Ag and the nanocomposites Ag NPs/cotton and Ag NPs/HBPAPES/cotton obtained by MVS were studied by powder X-ray diffraction. Figure 8 shows the PXRD pattern of the Ag powder prepared in the absence of cotton, using isopropanol as an organic reagent. All reflections observed in the pattern are assignable to the cubic phase of Ag metal (Fm-3 m space group) [45], with the lattice parameter *a* equal to 4.087 Å. The corresponding peaks are clearly seen in Figure 8 at the 2θ angle values of 38.1° (111), 44.3° (002), 64.5° (022), 77.4° (311) and 81.5° (222). An evaluation of the crystallite size (X-ray coherent scattering region) of silver particles was performed by the Scherrer method from the peaks broadening [46], using the K coefficient value of 0.89 as envisioned for spherical species. The size value of 21.1 nm was obtained, which indicates a nanoscale character of the particles synthesized by the mentioned method. 

PXRD patterns of the nanocomposites Ag NPs/cotton and Ag NPs/HBPAPES/cotton are shown in Figure 9 and Appendix A, respectively. As can be seen, the reflections of the support material [47] dominate in corresponding patterns. Apart from them, the only reflections of metallic silver are present in the 2θ > 36° region, as evidenced by the fit with the above-mentioned cubic phase of Ag accounting for the support reflections present in this region. Overall, the diffractograms of both composite materials are similar to each other. However, they differ significantly in the broadening of the silver phase reflections. Determination of the crystallite sizes of Ag nanoparticles has given the size of 14.5 nm for the Ag/cotton composite, while the size decreases drastically to 5.7 nm in case of the Ag NPs/HBPAPES/cotton composite.

The obtained results show that the cotton support favours a decrease in the size dimensions of the silver particles produced by MVS and that the most disperse phase of silver is stabilized in the presence of the hyperbranched polyorganoalkoxysiloxane nanogel coating. It was previously shown that gold nanoparticles obtained by the MVS, when interacting with alkylsilanes, lead to the formation of polymer structures that stabilize metal nanoparticles [48,49].

X-ray photoelectron spectroscopy was applied to study the surface of cotton modified with an Ag-containing organosilicon nanogel. For the analysis of an Ag NPs/HBPAPES/cotton hybrid nanocomposite, the use of controlled differential charging with the application of a bias voltage of different polarity U_b_ to the sample holder was used for the first time.

When using XPS for the analysis of non-conductive samples, a positive charge is usually accumulated on the surface due to photoelectron emission. This charge shifts the photoelectron peaks towards higher binding energies. Regions with different secondary electron emission coefficients (γ), work functions (φ) and conductive properties (ρ) are charged differently, which causes broadening and shift of the observed photoelectron spectra [50,51,52,53,54]. This phenomenon, which characterizes the inhomogeneity of the surface, is called differential charging. Partial charge compensation is carried out by a flux of secondary electrons emitted by the foil of the x-ray gun of a non-monochromatized photon source; it can be adjusted by applying a bias voltage (U_b_) to the sample holder. At U_b_ > 0, the electron flux to the sample increases, and the potential surface relief becomes even. The shape of the spectral line changes (as a rule, the peaks become narrower) and the spectrum shifts towards higher binding energies E_b_. If U_b_ < 0, the flux of electrons reaching the sample surface decreases, and the differential surface charging increases, which makes it possible to detect inhomogeneities on the sample surface. When U_b_ is applied, the photoelectron peaks recorded for conductive regions are shifted by U_b_, while the shift of peaks from non-conductive regions depends on γ, φ and ρ. Thus, this approach makes it possible to reveal phase differences in materials.

Appendix A shows survey XPS spectra of the cotton and Ag NPs/HBPAPES/cotton samples. Quantification data of the cotton and Ag NPs/HBPAPES/cotton samples presented in Appendix A show the absence of surface contaminations of the sample after the synthesis of the nanocomposite. Figure 10 shows the C 1s and O 1s photoelectron spectra of the cotton sample measured at different bias voltages. It was found that applying a negative and positive bias voltage of 7 V to the sample holder leads to a shift in the spectra by −0.2 and 2 eV, respectively. This is due to the fact that at U_b_ = 7 V, the flux of stray electrons to the sample surface increases, while at U_b_ = −7 V, it decreases. The shape of the spectra does not change, which may indicate the homogeneity of the original cotton. When superimposed, the spectra of C 1s and O 1s almost coincide, which also indicates the uniform composition of the cotton surface.

The C 1s and O 1s spectra of the cotton sample were fitted with four Gaussian peaks in accordance with reliable chemical shifts [55]. Figure 11 shows the C 1s and O 1s spectra measured at U_b_ = 7 V.

Appendix A shows their characteristics, from which it follows that impurity or defective carbon is observed in the C-C/C-H and C(O)O groups (C1 and C4 peaks), the relative intensities of which are the same and equal to 0.18. The relative fraction of impurity oxygen in the O 1s spectrum (C1 and C4 peaks) is 0.26.

After taking into account the surface charge, the coincidence of the spectra is preserved, which indicates the uniform composition of cotton. Similar studies at different bias voltages were performed for Ag NPs/HBPAPES/cotton.

Figure 12 shows the C 1s, O 1s, Si 2p, N 1s and Ag 3d spectra of the cotton (**1**) and Ag NPs/HBPAPES/cotton (**2**) samples measured at different U_b_. It is clearly seen that the spectral shifts of Ag NPs/HBPAPES/cotton under bias have become significantly larger in comparison with those of Ag NPs/HBPAPES/cotton, and there are significant changes in the spectral line shapes at U_b_ = −7 V. This indicates the inhomogeneity of the surface composition, namely, the presence of regions differing in γ, φ and ρ.

The fact that the energy intervals between the spectra measured at different U_b_ values increased significantly relative to those of cotton spectra indicates a noticeable increase in the electrical conductivity of the surface and a high degree of coating of the cotton surface with Ag NPs/HBPAPES/cotton.

When the C 1s, O 1s, N 1 s and Ag 3 d spectra measured at U_b_ = 7 V are shifted by −5.22 eV, they practically coincide with those measured at U_b_ = 0 V. A shift less than the bias voltage indicates non-metallic conduction. The shift of the spectra measured at U_b_ = −7 V by 5.22 eV leads to only a partial overlap of the spectra.

For further interpretation of the spectra, the surface charge was taken into account, for which the C-C/C-H state was identified in the C 1s spectra measured at U_b_ = 0 V and U_b_ = 7 V. From the behaviour of the spectra upon application of a bias voltage, which reflects a significant improvement in the conductivity of the near-surface region, it can be assumed that it is largely composed of aminosilane and silver. Thus, the main part of the C 1s spectra was approximated according to the chemical structure formula of silane, and the high-energy regions were approximated by the cotton spectrum (C-OH and O-C-O groups) and the C(O)O group, which is observed in the C 1s spectra of the cotton sample (Appendix A). The characteristics of the photoelectron peaks are listed in Appendix A. The relative concentrations of silane, cotton and C(O)O groups in the C 1s spectra (a) and (b) are 0.9, 0.06, 0.04 and 0.91, 0.06, 0.03, respectively.

The small relative concentration of cotton in the near-surface region may be related to either an almost complete coating of cotton with aminosilane with little unmodified surface or an inhomogeneous coating thickness that may be comparable to the sampling depth of the C 1s photoelectrons. These data, as well as the fact that the C 1s, O 1s, N 1 s and Ag 3 d spectra measured at U_b_ = 0 V and U_b_ = 7 V are almost identical, indicate vertical differential charging at U_b_ = −7 V caused by various thickness layers of aminosilane.

In the Si 2p spectra measured at U_b_ = 0 V and U_b_ = 7 V (Figure 13), the main peaks coincide, but an additional peak in the region of low binding energies is observed. The energy interval between the peaks is 2.88 eV. A smaller shift of the second peak indicates that this peak belongs to the region with lower electrical conductivity and possibly corresponds to the Si^4+^ state.

When fitting the spectra, the ratio between the widths of the C 1s and Si 2p_3/2_ peaks was chosen according to the data for poly(dimethyl siloxane) [50]; the first is 0.13 eV less than the second. Since the widths of the peaks corresponding to different silane groups are in the range of 1.5–1.91 eV, this indicates that the width of the peaks in the Si 2p spectrum cannot exceed 1.8 eV. Figure 13 shows fitting of the Si 2p spectra measured at U_b_ = 0 V and U_b_ = 7 V, with the same peak widths of 1.8 eV. The Si 2p_1/2_–Si 2p_3/2_ spin-orbit splitting is ~0.61 eV and Si 2p_3/2_/Si 2p_1/2_ branching ratio is 2. The fitting parameters for the Si 2p spectra are presented in Appendix A. The peaks at ~102.3 and 104 eV are attributed to silicon atoms in the aminosilane and Si^4+^ state, respectively. As noted above, the peak at 99.8 eV is attributed to the Si^4+^ state, despite the lowest binding energy.

An analysis of the N 1s spectra measured at U_b_ = 0 and U_b_ = 7 V also showed that they cannot be described by a single Gaussian peak. Figure 13 shows their approximations by two peaks with approximately the same widths, but with different energy intervals between them. According to the chemical structural formula, only one state should be present in the spectra of the Ag NPs/HBPAPES/cotton sample; therefore, one of the most possible reasons for the appearance of the second state can be differential charging. It is caused by the deposition of Ag NPs/HBPAPES, which led to a non-uniform charge distribution on the analysed surface. Their noticeable manifestations are shown in Figure 14, which shows the Si 2p, N 1 s, Ag 3d, C 1s and O 1s spectra measured at U_b_ = −7 V that have a more complex structure, indicating the presence in the Ag NPs/HBPAPES/cotton sample of at least two phases, which differ in electrical conductivity.

The shift in the spectra by 5.22 eV does not lead to the coincidence of either low-energy or high-energy regions or the main maxima. As in the case of pure cotton, when a negative bias voltage is applied, the C 1s and O 1s spectra shift by a smaller amount than in the case of a positive voltage. This is due to a decrease in the electron flux from the X-ray gun foil and a corresponding decrease in surface charge compensation degree. Based on the general patterns of behaviour of the spectra when a bias voltage is applied, the low-energy regions of the spectra should be assigned to regions with higher electrical conductivity, which, apparently, is associated with a higher concentration of silver nanoparticles and their more uniform distribution. In all spectra, the low-energy slope is less steep than the high-energy slope (Appendix A), which can be explained by the size distribution of Ag nanoparticles, which leads to a corresponding change in γ, φ and ρ.

The Ag 3d spectra of the Ag NPs/HBPAPES/cotton sample measured at U_b_ = 0 V (a) and U_b_= 7 V (b) (Appendix A) are characterized by 3d_5/2_–3d_3/2_ peaks at 367.7—373.7 eV and 367.6–373 eV, respectively, with the characteristic branching ratios of 0.67 and rather similar peak widths of ~2 eV. These binding energies indicate the state of Ag^2+^ state [56].

Thus, the following pseudo-layer structure of the near-surface region can be proposed, consisting of bulk cotton, the Ag NPs/HBPAPES/cotton interface layer and the Ag NPs/HBPAPES top layer.

Despite the simultaneous precipitation of aminosilane and Ag on cotton, the formation of Ag NPs/cotton and Ag NPs/HBPAPES interfaces cannot be ruled out. The fact that the proportion of the cotton signal in the C 1s spectrum is small indicates that there is no signal from the first layer and that the detected photoelectrons are mainly emitted from the interface and top layer. As the flux of stray electrons at U_b_ = −7 V decreases, the surface charge compensation occurs mainly in the uppermost layers of the sample, predominantly consisting of Ag NPs/HBPAPES and, to a lesser extent, at the interface layer. The presence of a two-peak structure, in the C 1s and O 1s spectra, for example, indicates a narrow interface.

To determine the relative proportion of the phase with low electrical conductivity in the spectra measured at U_b_ = −7 V (Figure 14), the high-energy regions are approximated by the spectra measured at U_b_ = 0 V. These proportions are: Si-0.43, N-0.45, Ag-0.35, C-0.38 and O-0.34, and the corresponding composition can be expressed with the formula C_28.2_O_6.3_N_1.2_Ag_0.2_Si_1.8_. The composition of other phases corresponds to the formula C_45.9_O_12.3_N_1.5_Ag_0.4_Si_2.3_. It should be noted that, in both cases, the Si/N ratios are close to 1.5 and 1.6, respectively, which indicates that the aminosilane structure is retained, while the Si/Ag ratio in the first case is 1.5 times higher than in the second. However, since the concentration of Ag in relation to aminosilane is higher in regions with low electrical conductivity than in regions with higher electrical conductivity, it can be concluded from this that the Ag NPs located in them are either larger or the oxide shell is thicker.

Thus, from a comparison of the data obtained by TEM, XRD and XPS, one can conclude that spherical silver nanoparticles have a “core-shell” structure. According to surface sensitive XPS data (sampling depth is ~5 nm), silver is in the Ag^2+^ state, while bulk sensitive XRD (sampling depth is in micrometre range) show Ag^0^ state. It means that the outer layer of Ag NPs consists of AgO, while bulk consists of Ag^0^.

The use of controlled differential charging made it possible to detect phases of different conductivity in the obtained material, induced by the formation of silver-containing organosilicon hybrids, the electronic state of which differs significantly from both the polymeric organosilicon matrix and the cotton substrate.

The data obtained made it possible to suggest the structure of the hybrid material: the basis is cotton, which provides strength characteristics and the fibrous–porous structure of which effectively stabilizes the silver-containing organosilicon nanogel. It is responsible for the physical or chemical sorption of the metal polymer on the polysaccharide. The upper layer of the material is almost completely formed from an organosilicon metal polymer, which, apparently, determines the antibacterial properties of the resulting composite. Of course, silver nanoparticles play a key role here, since other components of the system-cotton and organosilicon nanogels are biocompatible polymers, but do not have bacterial activity.

In this regard, the electronic state of silver nanoparticles and the role of aminosilanes in its formation are an important factor for understanding the biological activity of the system as a whole.

The creation of a new generation of medical materials based on a biocompatible natural polymer (cotton) and biologically active metal nanoparticles is one of the most relevant and intensively developed scientific directions [57,58].

Preliminary tests of the obtained nanocomposites showed high antimicrobial activity against Gram-positive bacteria *B. subtilis* ATCC 6633 and Gram-negative bacteria *E. coli* ATCC 25922. Ag NPs/HBPAPES/cotton and Ag NPs/HBPAPES/cotton-2 (re-impregnated sample) inhibited growth of *B. subtilis* ATCC 6633 and *E. coli* ATCC 25922; the size of the inhibition zones varied within 15 ± 1 and 30 ± 1 mm, respectively. The activity of the re-impregnated sample was higher than that of the amoxiclav control. As can be seen, the lowest activity was detected for *St. aureus* ATCC 25923 compared to antibiotic amoxiclav (Table 1).

Interestingly, moderate activity was also detected for fungi A. niger INA 00760 compared with control antifungal drug amphotericin B (Figure 15 and Table 1). As shown in Figure 15, the films exhibited weaker antifungal effect than antibacterial activity.

It has been proven that the resulting nanocomposites have a fairly pronounced antimicrobial activity against fungi and bacteria. Moreover, in the case of Gram-negative bacteria *E. coli* ATCC 25922, the antibacterial activity of composites was greater than that of the antibiotic. This makes the structure with Ag NPs/HBPAPES/cotton a prospect for the development of antibacterial coating.

## 3. Materials and Methods

### 3.1. Materials

Starting reagents were purchased from Acros Organics (Geel, Belgium) or Sigma-Aldrich Inc. (Saint Louis, MO, USA). All solvents were of reagent grade and dried and distilled before use according to standard procedures.

The ordinary medical gauze bandage (cotton) was used in the study, produced in the Republic of Belarus (State Standard 1172–93).

#### 3.1.1. Synthesis of Sodiumoxy-(3-aminopropyl)diethoxysilane (Figure 1)

The mixture of the (3-aminopropyl)triethoxysilane (0.05 mol, 11.07 g) and sodium hydroxide (0.05 mol, 2.0 g) was stirred in boiling THF (40 mL) for 20 min under argon. After cooling, the resulting solution was evaporated under reduced pressure and vacuumed at 1 mBar for 30 min at 50 °C. The product was obtained as colourless liquids with quantitative yields. 

#### 3.1.2. Synthesis of Hyperbranched Poly(3-aminopropyl)ethoxysiloxane (Figure 2)

To the solution of the previously obtained sodiumoxo(3-aminopropyl)diethoxysilane (0.022 mol, 4.74 g) in 20 mL of toluene at room temperature, an equimolar amount of acetic acid (0.022 mol, 1.32 g) was rapidly added. The solution was left for stirring overnight. The following day, the resulting solution was centrifuged, decanted from the precipitate, evaporated under reduced pressure and vacuumed at 1 mBar for 2 h at 50 °C. The product represented a colourless viscous liquid and was obtained in a quantitative yield.

#### 3.1.3. Synthesis of Nanocomposite Ag NPs/HBPAPES/Cotton

Ag nanoparticles were obtained by MVS, during which metal–organic reagent vapours were co-condensed in a vacuum of 10^−2^ Pa on the walls of a quartz reactor with a volume of 5 L cooled to 77K. Ag was evaporated by resistive heating from a tantalum boat in high vacuum with an organic reagent–isopropanol, which is supplied to the reactor via a separate line with a tap. In the process of their co-condensation, a solid matrix of co-condensate is formed on the reactor walls cooled by liquid nitrogen. At the end of the reaction, the cooling is removed, the reactor is filled with argon, the cryomatrix melts and the resulting organosole of metal nanoparticles was transferred through an in situ siphoning system to a flask containing an organosilicon polymer and stirred for 30 min. The resulting nanocolloid was later used to modify cotton.

The bandages (cotton) were cut into 6 by 6 mm squares and placed in the flask containing an organosol of silver nanoparticles (1.25 mg/mL), isopropanol and HBPAPES. After active mixing, the modified bandages (Ag NPs/HBPAPES/cotton) were dried in a vacuum unit to constant weight. One third of the total volume of Ag NPs/HBPAPES/cotton was re-impregnated with S-polymer organosol (Ag NPs/HBPAPES/cotton-2).

### 3.2. Methods

Solution phase NMR spectra were acquired using a Bruker Avance AV-300 (300 MHz for ^1^H; 77.5 MHz for ^13^C; 59.6 MHz for ^29^Si) spectrometer. Chemical shifts are reported in ppm and referred to the residual non-deuterated solvent frequencies (δ = 7.25 ppm) for ^1^H NMR; deuterated chloroform (δ = 77.00 ppm) for ^13^C NMR; tetramethylsilane (δ = 0.00 ppm) for ^29^Si NMR; and ammonia (δ = 0.00 ppm) for ^15^N NMR, sodium chloride (δ = 0.00 ppm) for ^23^Na NMR.

Mass spectra of high resolution were measured with the Bruker microtof II with an electrospray ionization device (ESI) (Germany).

FTIR spectra were measured with a Bruker “Equinox 55/S”. Potassium bromide permanent sealed liquid cells were used for measurements, solvent-CCl_4_.

Transmission electron microscope (TEM) images were performed with a transmission electron microscope LEO 912AB OMEGA, Zeiss (Oberkochen, Germany) at an acceleration voltage of 100 kV. Particle size distribution histograms were obtained from electron micrographs by manual calculation. The total number of particles is 200. The Gaussian function was approximated using the Sigma Plott (11 version) (Systat Software Inc., Richmond, CA, USA).

SEM images for samples placed on a 25 mm aluminum table and secured with a conductive carbon tape were obtained in the secondary electron mode at an accelerating voltage of 15 kV and medium vacuum mode on a Hitachi TM4000Plus desktop electron microscope equipped with an energy dispersive X-ray spectrometer (QUANTAX 75, Bruker).

Powder X-ray diffraction (PXRD) phase analysis was performed with a D8 Advance (Bruker AXS) diffractometer in the Bragg-Brentano focusing geometry using CuK radiation and an angular range of 5–90° with a step of 0.02^o^ and the scan rate of 0.5–2 deg min^−1^. The samples were placed on flat holders. Diffraction pattern profiles were fit using the TOPAS 5 program package (Bruker AXS, Karlsruhe, Germany).

X-ray photoelectron spectra were acquired with a XSAM-800 spectrometer using Al Kα (1486.6 eV) radiation at an operating power of 150 W of the X-ray tube. Survey and high-resolution spectra of appropriate core levels were recorded with step sizes of 1 and 0.1 eV, respectively. The samples were mounted on a sample holder with a two-sided adhesive tape, and the spectra were collected at room temperature. The base pressure in the analytical UHV chamber of the spectrometer during measurements did not exceed 1.33⋅10^−8^ Pa. The energy scale of the spectrometer was calibrated to provide the following values for reference samples (i.e., metal surfaces freshly cleaned by ion bombardment): Au 4f_7/2_–83.96 eV, Cu 2p_3/2_–932.62 eV, Ag 3d_5/2_–368.21 eV. The surface charge was taken into account according to the C-OH state identified in the C 1s spectrum of the cotton sample, to which a binding energy of 286.73 eV was assigned. In the case of the Ag NPs/HBPAPES/cotton sample, a binding energy of 285.0 eV was assigned to C-C/C-H state. After charge referencing, a Shirley-type background with inelastic losses was subtracted from the high-resolution spectra. The high-resolution photoelectron spectra were fitted with the sum of Gauss functions after charge referencing and Shirley background correction and satellite subtraction. Atomic surface concentrations were calculated from the high-resolution core level spectra using atomic sensitivity factors included in the software.

### 3.3. Estimation of DND Antimicrobial and Antifungal Activity

The antimicrobial activity Ag NPs/HBPAPES/cotton and Ag NPs/HBPAPES/cotton-2 (re-impregnated sample) were assessed by the agar diffusion method. Inhibition zones were measured manually using a digital calliper. Assays were performed three times in triplicate. Amphotericin B 40 µg («NII Pasteur», Saint Petersburg, Russia) and amoxiclav 10 µg («NII Pasteur», Saint Petersburg, Russia) were used as positive controls. The antibacterial activity was assessed with the following test strains: Gram-negative bacteria of *Escherichia coli* ATCC 25922; Gram-positive bacteria *Bacillus subtilis* ATCC 6633 and *Staphylococcus aureus* ATCC 25923. The antifungal activity was assessed with the fungi of the genus *Aspergillus*–*A. niger* INA 00760. The test culture of *B. subtilis* ATCC 6633 was grown on the Gause 2 medium (g/L): 2.5 tryptone (or 30 mL Hottinger broth), peptone-5, sodium chloride-5 and glucose-10, *Staphylococcus aureus* ATCC 25923 was grown on Müller–Hinton medium and *E. coli* ATCC 25922 was grown on the LB (tryptone soy agar). *A. niger* INA 00760 was grown on the PDA (potato-dextrose agar).

## 4. Conclusions

New hybrid materials with antibacterial activity, based on cotton materials widely used in medicine, modified with a silver-containing organosilicon polymer, were obtained using metal–vapour synthesis. It has been established that the use of a hyperbranched organosilicon nanogel contributes to the effective stabilization of Ag nanoparticles with an average size of 5.3 nm.

Analysis of the XPS data made it possible to describe the proposed structure of the hybrid material: its basis is cotton, on the surface of which a layer is formed responsible for the chemical and physical sorption of the silver-containing polymer. The top layer of the nanocomposite almost entirely consists of an organosilicon metal polymer.

XPS results showed that the Ag NPs/HBPAPES/cotton composite contained Ag^0^ and Ag^δ+^ states. In the nanocomposite, the metal nanoparticles have a “core-shell” structure, in which the “core” represents the Ag^0^ state, while the “shell” is the Ag^δ+^ one. The positive charge is apparently due to the interaction of the surface metal atoms with the functional groups of the polymer.

The nanocomposite Ag NPs/HBPAPES/cotton demonstrated antimicrobial action against the Gram-negative bacteria of *Escherichia coli* and Gram-positive bacteria *Bacillus subtilis.* Moreover, both forms show an efficacy against Gram-negative bacteria *Escherichia coli* approximately equal to the commercial antibiotic, amoxiclav.

Interestingly, activity was also detected for fungi *A. niger* INA 00760. Note that although only moderate activity was detected, this result can be determined by the low ability to diffuse on agar compared with a diffusivity of the control antibiotics. Thus, alternative materials with antifungal activity toward mold fungi are no less important than antibacterial ones.

Although the antibacterial effect of Ag is known, the new approach to the synthesis of hybrid metal–polymer systems presented in the work can make a significant contribution to the creation of a new generation of medical devices for potential clinical applications.

## Figures and Tables

**Figure 1 pharmaceutics-15-00809-f001:**
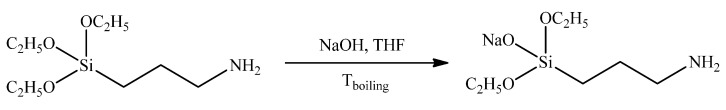
Synthesis of sodiumoxy-(3-aminopropyl)diethoxysilane.

**Figure 2 pharmaceutics-15-00809-f002:**
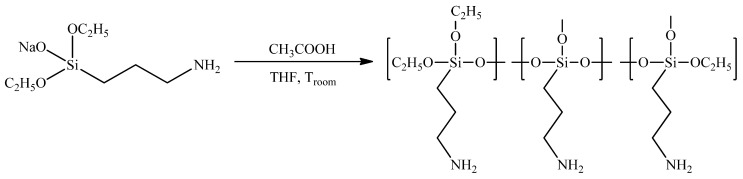
Synthesis of hyperbranched poly(3-aminopropyl)ethoxysiloxane.

**Figure 3 pharmaceutics-15-00809-f003:**
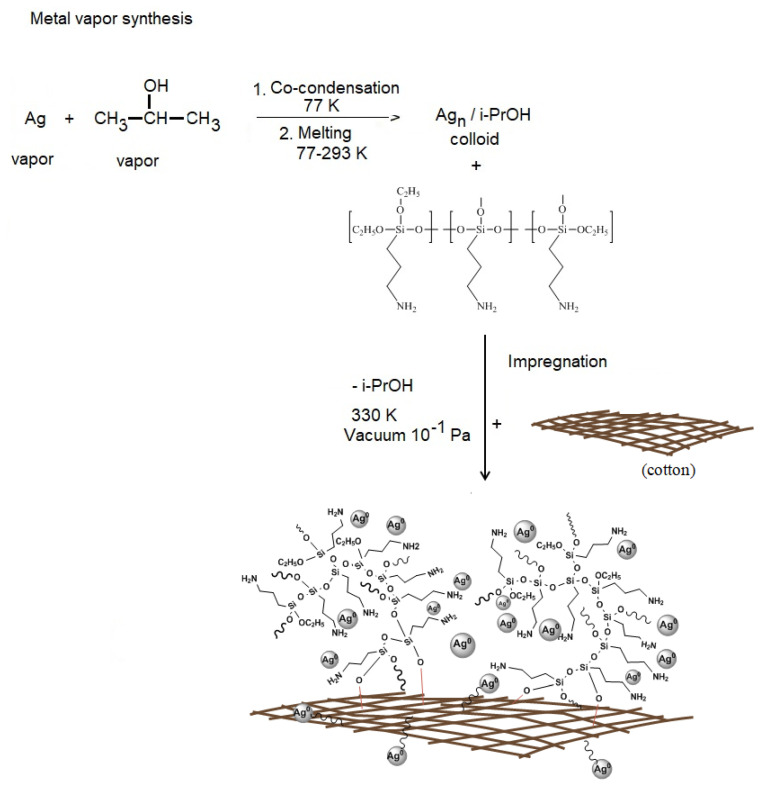
Scheme for obtaining nanocomposite Ag NPs/HBPAPES/cotton.

**Figure 4 pharmaceutics-15-00809-f004:**
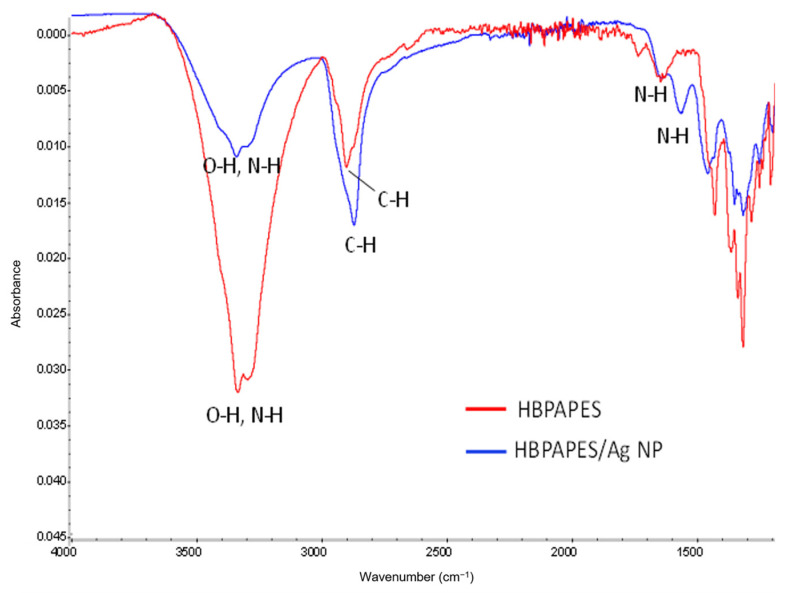
FTIR spectra of pure HBPAPES and HBPAPES with encapsulated Ag NPs.

**Figure 5 pharmaceutics-15-00809-f005:**
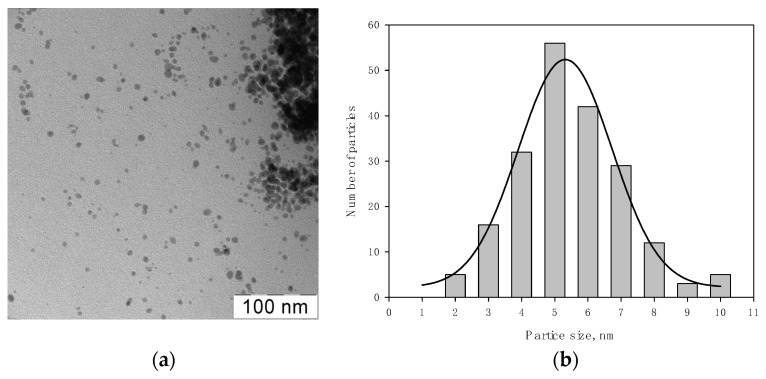
TEM micrographs of Ag NPs in a bright (**a**) field and the histogram of the distribution of Ag nanoparticles from the Ag NPs/2-propanol/HBPAPES system (**b**).

**Figure 6 pharmaceutics-15-00809-f006:**
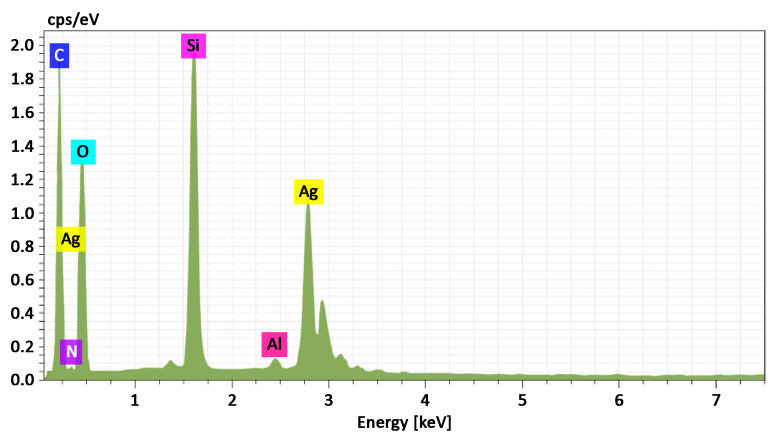
Energy-dispersive X-ray spectrum from Ag NPs/HBPAPES/cotton, mass. %. C 47.98; N 3.67; O 37.05; Si 7.72; Ag 3.42.

**Figure 7 pharmaceutics-15-00809-f007:**
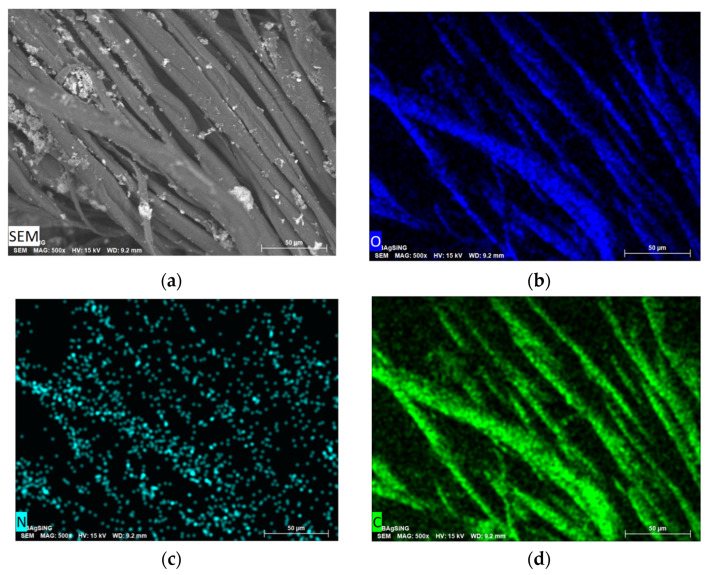
SEM image of the morphology of Ag NPs/HBPAPES/cotton (**a**) and the elemental distribution of O (**b**), N (**c**), C (**d**), Si (**e**) and Ag (**f**).

**Figure 8 pharmaceutics-15-00809-f008:**
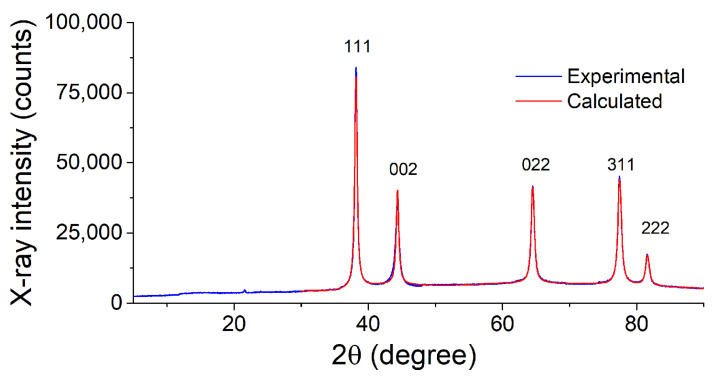
X-ray diffraction pattern of Ag NPs powder obtained by MVS and its fit.

**Figure 9 pharmaceutics-15-00809-f009:**
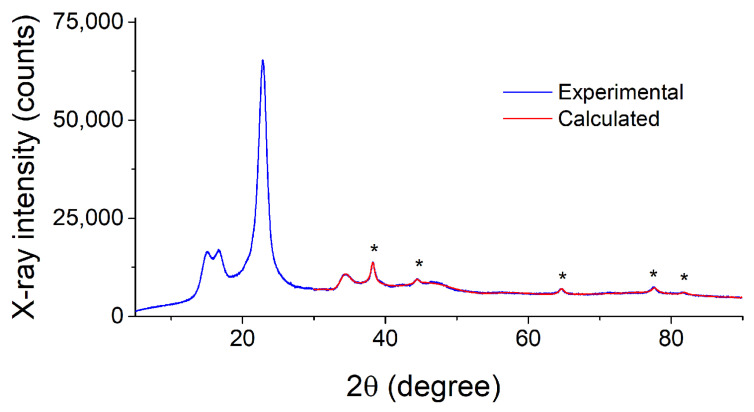
PXRD pattern of the Ag NPs/cotton composite and its fit, evidencing the presence of Ag nanoparticles. The reflections of metallic silver are marked by asterisks.

**Figure 10 pharmaceutics-15-00809-f010:**
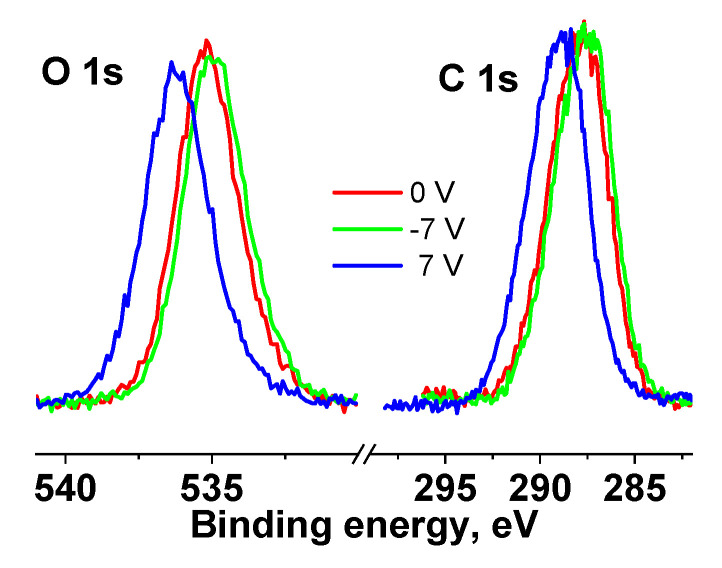
XPS C 1s and O 1s spectra of the cotton sample measured at different U_b_, the spectra are normalized by an area under the peak.

**Figure 11 pharmaceutics-15-00809-f011:**
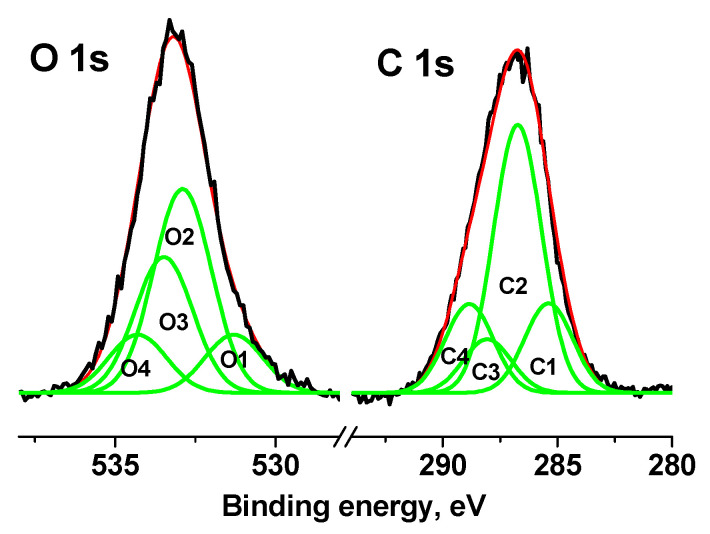
XPS C 1s and O 1s spectra of cotton sample measured at U_b_ = 7 V.

**Figure 12 pharmaceutics-15-00809-f012:**
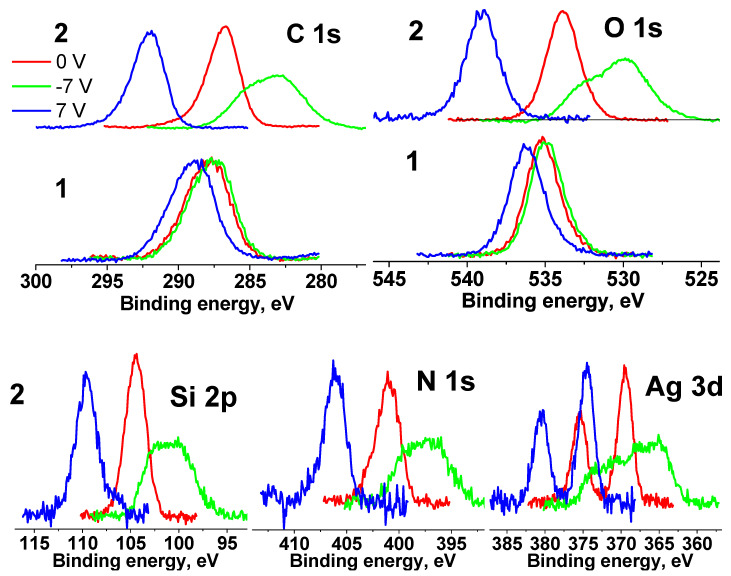
XPS C 1s, O 1s, Si 2p, N 1s and Ag 3d spectra of the cotton (**1**) and Ag NPs/HBPAPES/cotton (**2**) samples measured at different U_b_, the spectra are normalized by an area under the peak.

**Figure 13 pharmaceutics-15-00809-f013:**
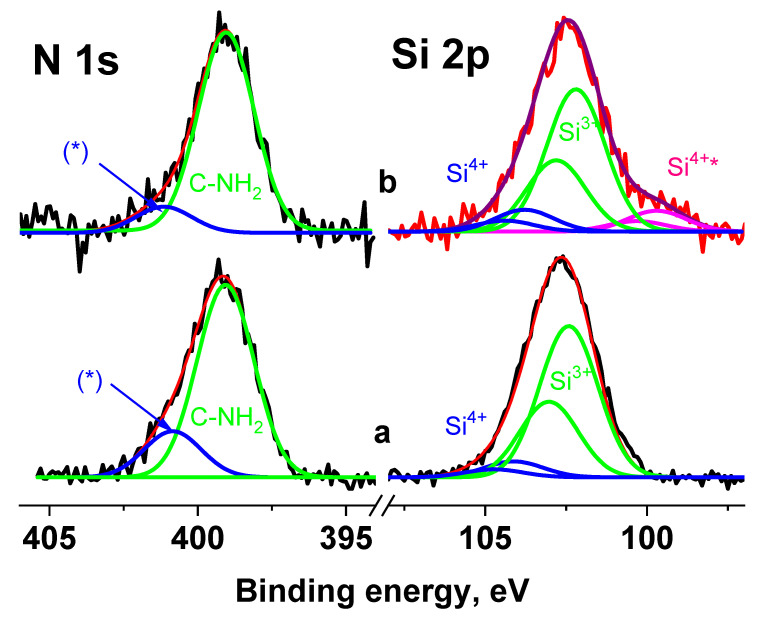
The Si 2p and N 1s spectra of the Ag NPs/HBPAPES/cotton sample measured at U_b_ = 0 V (**a**) and U_b_ = 7 V (**b**), *—differential charging.

**Figure 14 pharmaceutics-15-00809-f014:**
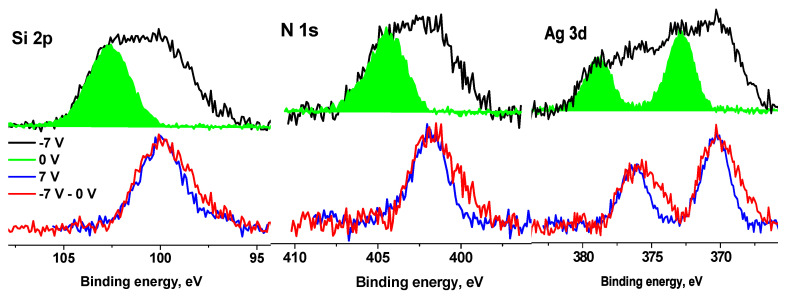
The XPS Si 2p, N 1s, Ag 3d, C 1s and O 1s spectra of the Ag NPs/HBPAPES sample measured at different U_b_.

**Figure 15 pharmaceutics-15-00809-f015:**
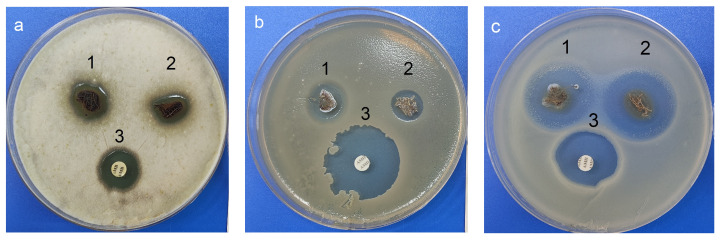
Antimicrobial activity: (**a**) **1**—Ag NPs/HBPAPES/cotton; **2**—Ag NPs/HBPAPES/cotton-2; **3**—amphotericin B; (**b**,**c**) **1**—Ag NPs/HBPAPES/cotton; **2**—Ag NPs/HBPAPES/cotton-2; **3**—amoxiclav. *A. niger* INA 00760, *B. subtilis* ATCC 6633, *E. coli* ATCC 25922.

**Table 1 pharmaceutics-15-00809-t001:** Antimicrobial activity of Ag NPs/HBPAPES/cotton and Ag NPs/HBPAPES/cotton-2.

Sample	Zone of Inhibition, mm			
	*B. subtilis* ATCC 6633	*S. aureus* ATCC 25923	*E. coli* ATCC 25922	*A. niger* INA 00760
Ag NPs/HBPAPES/cotton	15 ± 0.4	7 ± 1	26 ± 1	13 ± 1
Ag NPs/HBPAPES/cotton-2 (re-impregnated sample)	15 ± 1	8 ± 0.1	30 ± 1	13 ± 0.7
Amoxiclav 10 µg	29 ± 0.3	25 ± 0.3	25 ± 0.1	* nt
Amphotericin B 40 µg	* nt	* nt	* nt	15 ± 0.6

* nt—non-tested.

## Data Availability

Not applicable.

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
