# Peer review of "Hybrid Materials with Antimicrobial Properties Based on Hyperbranched Polyaminopropylalkoxysiloxanes Embedded with Ag Nanoparticles"

_pharmaceutics, 2023, doi:10.3390/pharmaceutics15030809_

Round 1

Reviewer 1 Report (Previous Reviewer 4)

The authors have corrected and rewritten most of the things that I required. Although, I am not satisfied with some of the answers, as the authors should have considered the advice. I will accept the work without additional changes due to the quality of the presented results. The technique used should be based on theoretical knowledge, and the author must know how to show the results well so that they are clear and do not burden the reader.

Reviewer 2 Report (Previous Reviewer 3)

The authors have now revised their manuscript carefully.

Reviewer 3 Report (Previous Reviewer 2)

Thank you very much for making the suggested corrections.

This manuscript is a resubmission of an earlier submission. The following is a list of the peer review reports and author responses from that submission.

Round 1

Reviewer 1 Report

The following points were observed for correction.

1.       Line no: 551, In table no:2:  B. subtilis TCC 6633 must be as ATCC 6633

2.       In table control antimicrobial disc concentration can be included.

3.       In line no: 655, 3.3. Estimation of DND Antimicrobial and Antifungal Activity

There is a contradictory need to be addressed and corrected according to the figure 21 and result.

In results section for  INA 00760 A. niger, the control discs used was Amphotericin-B  and for ATCC 25922 E. coli and  ATCC 6633 B. subtilis  ampicillin was used.

But in the methodology for A. niger Amphotericin B 40 µg and Nystatin 80 µg discs used as control disc and for the bacteria Amoxiclav 10 µg were used as positive control.

Authors are strongly recommended to revise thoroughly in the entire result section of the manuscript to avoid such typographical error

4.       Line no: 684 and 685 In conclusion:  The authors have mentioned as “The nanocomposite Ag NPs/HBPAPES/cotton demonstrated antimicrobial action against the gram-negative bacteria of Escherichia coli and gram-positive bacteria Bacillus subtilis.  

Why not included the A. niger  even though there was moderate inhibitory activity as mentioned in the result section line no 541. (In-vitro study no one can expect huge antifungal inhibition, Ref the standard drug Amphotericin-B zone of inhibition). Here with recommend to add the antifungal activity in conclusion.

5.       Why the authors have chosen B. subtilis in this study since it’s not a major human pathogen/ not a major drug resistant pathogen as they claimed in the introduction part.  Why they have not chosen the Staphylococcus sp or Pseudomonas aeruginosa.   They can justify in the discussion or recommended to add study limitations.

Reviewer 2 Report

COMMENTS TO THE ARTICLE: “Hybrid materials with antimicrobial properties based on hyperbranched polyaminopropylalkoxysiloxanes embedded with Ag nanoparticles”

This article deals with the synthesis of a new hybrid material based on Ag nanoparticles stabilized in hyperbranched polymer. Ag nanoparticles are obtained by metal vapour synthesis where there is a simultaneous evaporation and condensation of metal and organic ligand at temperatures of liquid nitrogen and under vacuum conditions. After that, nanoparticles are dispersed in isopropanol, mixed with hyperbranched polymer, and impregnated on cotton to obtain the Ag/hyperbranched/cotton nanocomposite.

The material was characterized by different experimental techniques like, among others, FTIR, TEM, PDRX, and mainly by XPS.

Even the article is interesting, some corrections must be carry out

First, in page 2 there is a repeated paragraph, lines 71-75. After modification, check the reference number at the end of the paragraph.

Figure 4 on page 7 related to FTIR spectra is impossible to see.

Figure 5 related to TEM micrographs and histogram of the distribution. Do you think that really the average size is 5.9 nm? On the other hand, how do you measure the size of the particles because most of them are agglomerated? Which software programme was used? Really, I plotted that data and fitted to a Gaussian curve and the average that I obtained was below 5 nm. Please, check the results.

Figure 6 on page 8 related to EDX. Please, in order to be better observed and analysed, and as after 6 keV there are no any peaks, expand the x-axis scale, I mean, the scale of x-axis should be between 0 and 6 keV. Then, peaks will be better seen.

In order to reduce the number of figures, maybe, figure 9 and 10 should be plotted in only one figure because are very similar.

XPS analysis is, in my opinion, too long. Must be reduced.

Page 11, lines 295-310 authors mentioned that when the analysis is carried out in non-conductive samples, a positive charge is accumulated on the surface… Also, authors presented a good description this phenomenon. I do not know if such description as well as parameters (emission coefficients, work functions, etc.) was developed by the authors or was obtained from other scientific article. Please, introduce any reference or can be considered as plagiarism.

Figures 11, 12,13, 14, 15, 16, 17, 18, 19 and 20 should be concentrated and present only 2 or 3 figures. On the other hand, y-axis label presented in such figures is Intensity, a. u.. However, there is a value in such axis; I mean, from 0 to 2, 0 to 4, etc. If the intensity is a.u. no number have to be in the y-axis.

Figure 14 shows the XPS analyses on cotton and nanocomposite. Authors named (1) and (2). However, in figure 13, for example, different plots are named (a) (b), (c)… How authors does not follow the same nomenclature in Figure 14?. Please, follow the same rule for all figures.

About the errors presented in line 539 related to the inhibition zones. The number of decimals of the obtained value depend on the first significant digit of the error. For example, in this case, authors indicated that the size of the inhibition zone is 15.33±1.18 mm. This value must be expressed as 15±1 because the first significant digit of the error is 1. Then the measured value can not be expressed with decimals. If the obtained value is, for example, 15.234 and the error 0.035, due to the firs digit of the error is 0.03, the value must be written as 15.23±0.03. Two decimals because the first significant digit of the error is the second decimal. Please, modify data of Table 2 following these procedures.

In the method section, authors indicated that they use different infrared spectroscopy techniques like DRIFT, ATR, Kubelka-Munk transformation, etc., However, only a FTIR mention is carried out in figure 4 but authors did no mention the used technique. Also, in the supplementary material there is no any DRIF FT-IR spectrum or Kubelka-Munk transformation. Then, if there is no results, please remove or modify such text form the article.

Finally, in relation to antimicrobial activity, this section is very short and maybe more results are needed.

In my opinion, the title of the article, even some of antimicrobial properties are presented, is mainly focused on the surface characterization by XPS. Then, maybe, should be send a journal of surface characterization, for example.

Reviewer 3 Report

The article reported the synthetic method of AgNPs by using metal vapor synthesis (MVS) and incorporated into the polymer matrix using metal-containing organosol. The article have some innovation, but there are many serious problems in layout and writing. This will make readers feel confused and uncomfortable. In my opinion, the current manuscript is difficult to be accepted and published by magazines. The details are presented as follows:

1. There are a lot of references in the introduction section, but few in the results and discussion section, and there is a lack of comparison and discussion of peer research. So, It is suggested to strengthen the literature comparison of the results and discussion.

2. A large number of words and pictures are used to characterize materials, such as TEM, XRD, and XPS. But the results are too scattered and the pictures are too redundant. It is suggested that the author put the unimportant data in the attachment, summarize and integrate the experimental data, and extract the theme according to the experimental results and conduct in-depth discussion.

3. The text in lines 66-70 and 71-75 is repeated.

4. Figure 4 is incomplete.

5. The discussion part of XRD is too long and too much content is meaningless. It is suggested to delete it.

6. Figure 13 lacks notes on a-f, and its results and discussion are not clear enough. The small picture in Figure 13-20 is too messy. It is recommended to synthesize a picture and arrange it neatly.

Reviewer 4 Report

The manuscript “Hybrid materials with antimicrobial properties based on hyperbranched polyaminopropylalkoxysiloxanes embedded with Ag nanoparticles” is suitable for publication in Pharmaceuticals

The authors prepared novel hybrid materials based on Ag nanoparticles stabilized by polyaminopropylalkoxysiloxane hyperbranched polymer matrix. Characterization of material was done with good technic, and also antimicrobial properties were examined.

Specific comments:

  1. Figure 4. Something is wrong with this picture. There aren’t any lines for FTIR spectra, only marked areas. Please, check and resubmit again.
  2. Figure 5. The author should give a picture of Ag nanoparticles with higher magnifications.
  3. Also, Fig 5 and 6 should be merged.
  4. Figures 8., 9., and 10.: Experimental and Calculated graphs should be separated into two graphs. Or they can stay in one graph, but one should be multiplied, and on abscissa should put arbitrary units.
  5. XPS measurement on 10 pictures is too much. Please, take only important results and present them there.

I believe that it should be published after MAJOR revision.